# Clinical Profiles and Socio-Demographic Characteristics of Adults with Specific Learning Disorder in Northern Greece

**DOI:** 10.3390/brainsci11050602

**Published:** 2021-05-08

**Authors:** Eleni Bonti, Sofia Giannoglou, Marianthi Georgitsi, Maria Sofologi, Georgia-Nektaria Porfyri, Artemis Mousioni, Anastasia Konsta, Paraskevi Tatsiopoulou, Afroditi Kamari, Sofia Vavetsi, Ioannis Diakogiannis

**Affiliations:** 1First Psychiatric Clinic, School of Medicine, Faculty of Health Sciences, Aristotle University of Thessaloni-ki, “Papageorgiou” General Hospital, Ring Road Thessaloniki, N. Efkarpia, 54603 Thessaloniki, Greece; sophiegianno@gmail.com (S.G.); geoporfyri@hotmail.fr (G.-N.P.); arta-4@windowslive.com (A.M.); konstaa@auth.gr (A.K.); vivitatsiopoulou@yahoo.gr (P.T.); afrkamari@yahoo.gr (A.K.); svavetsi@sch.gr (S.V.); idiakogiannis@auth.gr (I.D.); 2Department of Education, School of Education, University of Nicosia, 2417 Nicosia, Cyprus; 31st Laboratory of Medical Biology-Genetics, Faculty of Health Sciences, School of Medicine, Aristotle University of Thessaloniki, 54124 Thessaloniki, Greece; margeorgitsi@auth.gr; 4Psychology Laboratory, Department of Early Childhood Education, School of Education, University of Ioannina, 45100 Ioannina, Greece; m.sofologi@uoi.gr; 5Institute of Humanities and Social Sciences, University Research Centre of Ioannina (U.R.C.I.), 45100 Ioannina, Greece

**Keywords:** specific learning disorder, adults, clinical profiles, socio-demographic characteristics

## Abstract

The manifestation of Specific Learning Disorder (SLD) during adulthood is one of the least examined research areas among the relevant literature. Therefore, the adult population with SLD is considered a “rare” and “unique” population of major scientific interest. The aim of the current study was to investigate, describe, and analyze the clinical, academic, and socio-demographic characteristics, and other everyday functioning life-skills of adults with SLD, in an attempt to shed more light on this limited field of research. The overall sample consisted of 318 adults, who were assessed for possible SLD. The diagnostic procedure included self-report records (clinical interview), psychometric/cognitive, and learning assessments. The main finding of the study was that SLD, even during adulthood, continues to affect the individuals’ well-being and functionality in all of their life domains. There is an ongoing struggle of this population to obtain academic qualifications in order to gain vocational rehabilitation, as well as a difficulty to create a family, possibly resulting from their unstable occupational status, their financial insecurity, and the emotional/self-esteem issues they usually encounter, due to their ongoing learning problems. Moreover, the various interpersonal characteristics, the comorbidity issues, and the different developmental backgrounds observed in the clinical, academic, personal, social, and occupational profiles of the participants, highlight the enormous heterogeneity and the continuum that characterizes SLD during adulthood. We conclude that there is an imperative need for further research and the construction of more sufficient tools for the assessment and diagnosis of SLD during adulthood, which will take into account the developmental challenges and milestones in a series of domains, in order to assist this “vulnerable” population with their life struggles.

## 1. Introduction

### 1.1. Specific Learning Disorder (SLD) during Adulthood

According to the National Joint Committee on Learning Disabilities (NJCLD), Specific Learning Disorder (SLD) is a general term that refers to a heterogeneous group of disorders [1,2], which may involve difficulties in reading (also known as dyslexia), written expression (dysgraphia), and/or mathematics (dyscalculia) [3]. SLD is characterized by difficulties with one’s ability to process, organize, and retain verbal or nonverbal information [2,3].

The new APA DSM-5 describes SLD as a neurodevelopmental disorder with a biological origin, which includes an interaction of genetic, epigenetic, and environmental factors [1].

Symptoms of SLD are usually detected during the early school age, where students show a learning profile that is qualitatively lower than their chronological and mental age. However, in some cases, difficulties may become obvious at a later age, i.e., when the academic demands increase [2]. Therefore, in adults, persistent difficulty refers to ongoing difficulties in literacy or numeracy skills that manifest during childhood or adolescence, as indicated by cumulative evidence from school reports, evaluated portfolios of work, or previous assessments.

Symptomatology changes through different age periods, whereas, in several cases, comorbidity issues with other disorders make differential diagnosis an even more complicated task [4]. More specifically, during adulthood, SLD seems to affect the academic, vocational, social, and emotional development of the individual [5]. Low academic skills interfere with occupational performance or everyday activities requiring those skills. As a result, avoidance of activities that require academic skills is a typical clinical indicator during adulthood [4,6].

Despite these difficulties, many adults diagnosed with SLD as children seem to continue with post-secondary education, with the rate of university students with dyslexia in Greece reaching approximately 16% [7]. The only provision by the relevant legislation in Greece, for those formally diagnosed with SLD, is the exclusion from written exams during formal schooling [3], whereas in postsecondary education and during adulthood, the law is unclear regarding any kind of provision. According to the National Longitudinal Transition Study 2 (NLTS-2), 67% of adults with SLD were admitted to higher education, eight years after graduating from high school, but only 41% of them managed to graduate [8].

In a survey of 89 university students and graduates in Greece, many participants reported that the main problems they encountered during their studies concerned their concentration, the organization of their thinking and memory skills, as well as with the ability to effectively organize their time, with most of them stating that they delayed graduating [9].

As opposed to the school years, the “adult with SLD experience” does not constrain in education, since the disorder affects several other areas of their everyday life. In many cases, they claim having “left behind” their school years, possibly as an emotional reaction to their ongoing difficulties. Therefore, since SLD are now “invisible”, the adult has the choice to either reveal his learning problems or to conceal or mask them, and therefore, face the relevant consequences [10].

People with SLD usually struggle to find or to maintain a job, while the noises from the environment can distract them, creating an additional difficulty. In fact, 85% of these people do not reveal their problem to their employer by fear of rejection or dismissal [11]. According to Cortiella and Horowitz [2], only 19% reported to their employers that they have SLD and only 5% of them received appropriate support, in their workplace.

Geary’s research in the UK had shown that reading difficulties caused barriers to employment and reduced payment arrangements, while difficulties in mathematics had an even more serious impact on these individuals. Both men and women were less likely to find a full-time job, were mainly engaged in some manual work with low wages and with minor chances of any promotion, and experienced frequent periods of unemployment [12].

A study conducted in the USA investigated the employment status of 500 graduates with learning disabilities from three universities. These people were mainly employed in the business sector, while a significant percentage participated in education and health professions. The percentage of people who worked, in terms of their income, was comparable to that of the general population, in contrast with individuals with learning disabilities, who did not obtain a degree, indicating that university education provides favorable prospects for employment in a job. Unemployment rates in this group were around 5%, which corresponded to the rates in the general population [13]. Similarly, these findings were consistent with another study in which young people, after graduating from a university, were able to find full-time employment with a salary commensurate with the general population [14]. The young people who managed to secure a satisfactory job were those with significant mathematical skills and whose parents were actively involved in their education [14].

In general, according to Kaye’s survey, 46% of adults with SLD were employed, while 8% were unemployed, with the remaining 46% stating that they were not in the workforce [15]. Overall, it is worth noting that the employment rate of adults with SLD decreased from 55% (2005) to 46% in 2010 [15]. There are no further up-to-date data on the percentage of employed adults with SLD.

The lives of adults with SLD are affected in several ways. In their daily lives, they experience difficulties related to the management of time, ability to retain their concentration during a task or a discussion [16], and memory [17]. In addition, these adults often face several “misunderstandings”, are stigmatized, and become socially isolated, usually as a result of their low self-esteem [10].

The low self-esteem of adults with SLD seems to establish during the first school years, as shown in a study by Deacon, McDonald, and Donaghue [18]. Participants described their adolescence as traumatic in the school context and described themselves as isolated and inadequate. In many cases, the individuals also mentioned that, at the end of compulsory education schooling, they were not able to follow a career [18].

Many people with SLD, during their first school years, as well as during their adolescence, experience traumas and stigmatization. Depression and stress are very common, and as adults, these individuals realize the guiding role of the school environment in their subsequent success. They often refer to their life path as a course of “recovery-development” despite traumatic experiences, denying the identity of the “dyslexic-incompetent” person and using different methods, strategies, and skills, to better understand dyslexia [19].

Considering all the above, the personal and wider social relations of people with SLD are usually limited and/or dysfunctional. These people are sometimes excluded from social activities, mainly due to the tendency of being marginalized by others around them [20]. A person with SLD who is confronted with this “hostile” attitude by society, even from his early school years, as an adult, remains unfamiliar with normal social interaction; as a result, he often approaches others in aggressive or inappropriate ways. Hence, social exclusion begins in childhood and continues indefinitely.

Regarding the overall quality of life of adults with SLD, in terms of autonomy, functionality, and decision-making, qualitative studies have shown that the majority encounters difficulties regarding their successful adjustment in everyday life situations, interpersonal and social relationships, parenthood, as well as on an occupational level [4,14].

Usually, these people feel confident and emotionally stable only in their family environment, so they are reluctant to any changes, such as moving out from their family home. As a result, a significant proportion of adults with SLD continue to live with their parents and family, on whom they usually depend financially. According to a survey of young adults with SLD in Scotland, only 10% of them lived independently on their own, without help from a member or a home-based specialist [21]. The uncertainty of finding a job, which does not allow these people to perceive themselves as ‘adults’, leads to a reluctance as regards the prospect of starting their own family, due to financial difficulties. In fact, the same survey revealed that over 95% of the participants did not have a partner or spouse. This fact undoubtedly leads to lower self-esteem and a tendency to depression due to loneliness [21].

In general, adults with SLD have a strong desire to commence a romantic relationship; however, most of them emphasize the need to receive support from their family and social environment in order to start or manage to maintain these relationships. According to their statements, this support should include the acquisition of their own space and unimpeded access to means of transport [22].

At the same time, according to the qualitative research of Bond and Hurst [23], nine adults with SLD described the demands of everyday life as a constant struggle, in terms of their safety, health, and autonomy. They generally reported that they maintain family, friendship, and love relationships that sometimes support them financially, while at the same time, they admitted that most of their friends were also adults with SLD [23]. Finally, they stressed their desire for building more stable relationships with their close social environment, without being financially supported, in order to be able to manage issues that directly concern them, thus taking full control of their lives [23].

Other studies indicate that during adult life, there are different age periods (young adult, middle-aged, elder), in each of which the priorities differ. This fact also differentiates the needs of adults with SLD during the separate periods of adulthood (e.g., studies, family, vocational rehabilitation, social life, etc.) [4,24].

Learning disabilities affect a significant percentage of the population. As mentioned above, people who experience some form of learning disability, often suffer from low self-esteem, they tend to avoid setting high goals for life, and finally, struggle to find a job or maintain their social relationships. Thus, early diagnosis is crucial for proper intervention planning but can also serve as a protective factor for later low self-esteem and other emotional difficulties [2]. In addition, the family environment seems to play a key role, as the harmonious parent–child relationship positively contributes to the reduction of symptoms related to the child’s learning problems and improves their overall quality of life [25].

### 1.2. Adults with SLD: A Rare Population

As Sharfi and Rosenblum [14] state, there is no formal up-to-date information about the prevalence of SLD among adults internationally, whilst there is a very limited number of studies that investigate both academic and/or other life areas of this population, worldwide. The research body of information on SLD focuses on childhood while research in SLD during adulthood is limited within the last three decades [5,26,27]. More specifically, the existing studies mainly focus on the academic factor, while failing to investigate other life domains of this population, such as their future academic and occupational progress, their levels of participation in everyday and social activities, their interpersonal and social relationships, their family status, etc. [28,29]. According to Gerber [4], this research “gap” is mainly due to the fact that there is no common conceptual model regarding how to investigate the adult years of people with SLD.

Additional factors affecting the concept of SLD during adulthood, in total, are the following:

An adult-specific definition for SLD does not exist. At the same time, there is a wide range of functioning among individuals with SLD—from highly successful to moderately successful, to those who are either marginally adjusted and/or totally dependent on others. Thus, by nature, adults with SLD represent a continuum of severity, ranging from borderline or low average intelligence to superior intelligence. Complementary to the issue of severity is a range of adaptive behaviors that can have implications for daily functioning and social skills, which must be utilized consistently and effectively in numerous adult contexts.

Comorbidity issues are also part of the “adult with SLD experience”. It is not uncommon for adults with SLD to also manifest attention deficit disorder with or without hyperactivity (ADD/ADHD), anxiety, depression, personality disorders, and/or age-related conditions [30].

In addition, there is a wide array of inter-individual differences. Developmental challenges and milestones in a series of domains provide a matrix for adult functioning. Those domains vary depending on adult theorists, but they typically include employment, family, personal-social, and so on. Finally, even less is known about the reasons adults decide to refer for learning assessment or about the possible relations between these reasons and other factors, such as gender and type of SLD [31].

Generalizing, it becomes clear that the overall impact of SLD on adulthood can be very complicated.

Despite the limited body of literature in the field, it is obvious that SLD during the adult years can be extremely challenging in terms of diagnosis and manifestation of clinical characteristics across the individual’s life domains. Each year, large numbers of students with undiagnosed SLD leave high school and begin their adult lives facing a wide variety of challenges, which often lead to a broad array of negative outcomes, reflected in almost every area of their everyday life.

Especially during the last decade, a considerable number of adults refer to diagnostic centers, seeking help for their learning difficulties. The Outpatient State Diagnostic Department for Learning Difficulties (OSDDL) at the First Psychiatric Clinic of “Papageorgiou” General Hospital of Thessaloniki is one of the few State-certified diagnostic centers for the learning assessment of individuals over 18 years of age in Greece. More specifically, between 2012 and 2018, our department assessed over 350 adults with possible SLD.

Considering all the above, we consider adults with SLD as a “rare” and “unique” population of major scientific interest. Hence, the purpose of the present study was to shed more light and add valuable information to an internationally challenging area of research, i.e., the overall manifestation of SLD during adulthood. More specifically, the study aimed to present the clinical profiles, demographics, and other characteristics of this population, in several life domains. The range of data collected included variables, such as age, sex, referral request, intellectual level, the outcome of diagnosis, and other individual characteristics related to the academic-educational, professional, personal, and social level of the participants. Correlations between these variables were also considered.

In short, the impetus for this study was mainly the lack of relevant research data regarding adults with SLD in Greece and internationally, whereas the singularity of the particular research lies in the scarcity and “uniqueness” of this population.

## 2. Method

### 2.1. Research Methodology

#### 2.1.1. Participants—Inclusion and Exclusion Criteria

A total of 318 adults, aged from 18 to 56 years, were included in the sample, who visited the Outpatient State Diagnostic Department for Learning Difficulties (OSDDL) at the First Psychiatric Clinic of “Papageorgiou” General Hospital of Thessaloniki, from 2012 to 2018, requesting learning assessment (diagnosis for SLD). Specifically, the sample consisted of 198 males and 122 females from different parts of Greece. All of them were native Greek speakers.

Inclusion criteria: (1) older than 18 years and up to 56 years, (2) Greek nationality and Greek as a native language, (3) typical clinical profile for SLD diagnosis (diagnostic criteria of SLD provided by DSM-5 [1]), (4) mental capacity scored > 70 as measured by WAIS III or IV.

Exclusion criteria: (1) age older than 56 years due to possible cognitive decline, (2) a history of neurological or psychiatric disorders, as well as other serious medical conditions that might affect participants’ neuropsychological performance (e.g., stroke, epilepsy), (3) a history of neurodevelopmental disorders (e.g., mental retardation), (4) sensory impairments that might significantly interfere with cognitive testing.

#### 2.1.2. Procedure—Data Collection

Data included demographics, referral request, age, sex, mental level, the outcome of diagnosis, individual characteristics in relation to educational and/or professional level, as well as personal and social development of the participants.

Specifically, the diagnostic procedure included:

*Written informed consent:* Consent was obtained from all participants assessed in our Department.

*Confidential self-report records:* These were collected from adults with SLD through a clinical interview to obtain background information, social history, developmental history, (e.g., academic history/school reports, medical reports), and demographic data (e.g., personal, social, occupational reports). Data were collected through open and closed questionnaires.

The process of collecting background information from the participants was based on the DSM-5 indications and led to the construction of a self-reported questionnaire designed to collect as much as possible information about the overall history of the participants. This questionnaire is due to be standardized.

*Learning assessment:* Regarding the assessment of reading, writing, and mathematical skills, an assessment battery was used, which was released in 2013 and is also pending to be standardized.

The particular assessment provided us with a complete picture of the adults’ learning profiles. It is adapted to the Greek language, while its utilization on a large number of adults creates the prospect of standardization. Moreover, within the broader psychiatric department of the hospital, no other appropriate clinical learning and psychometric assessments for adults are available.

This assessment battery consists of several tasks evaluating basic, non-curriculum-based academic skills in the areas of literacy, language, and mathematics, which provide a full, sufficient, and clear picture of the different specific academic area skills of the adults being evaluated, within a short period of time. Each of the tasks assesses frequency or the level at which difficulties were detected and is scored in a Likert scale (0 = none or very rare, 1 = quite often, 2 = very often or systematically) [3].

More specifically, learning assessment consisted of the following tasks:

Assessment of reading skills. (1) Decoding skills: word-attack skills, errors (such as substitutions, omissions, inversions, insertions, etc.), line skipping, finger-pointing, hesitations, repetition of syllables/words/phrases, acknowledgment of punctuation, decoding of pseudowords, speed, rhythm, and expression. (2) Reading Comprehension: answering questions about the text, providing titles and subtitles, etc. (3) Phonological awareness: analysis and synthesis (phonemic segmentation) of syllables/letters containing complex consonant blends, digraphs, and other special letter combinations; counting of words within a sentence or syllables/letters within a word; and other phonological tasks (e.g., adding or omitting a letter in order to produce a new word, etc.)

Assessment of written language skills. Evaluation of handwriting, spelling, use of punctuation, structure, content, and linguistic errors (e.g., morphological, lexical, syntactic, and stylistic). Greek is considered a semi-transparent language since there is a substantial grapheme-phoneme correspondence. Therefore, a large proportion of words written phonetically are also orthographically correct.

Assessment of mathematical skills. (1) Calculation skills: performing operations (addition, subtraction, multiplication, and division), using alternative techniques. (2) Reasoning skills: understanding the text in the word problem, identifying keywords that lead to appropriate operation, following problem-solving steps, describing their reasoning.

Assessment of visual and auditory memory skills. The adult sees/reads and hears specific images, words, and sentences which (s)he is asked to recall or reproduce.

*Clinical and psychometric assessment:* Administration of the Wechsler Adult Intelligence Scale-III and IV psychometric tool (WAIS-III and IV) [32,33,34,35,36]. WAIS is necessary for the examination of the mental level of the participants, because according to DSM-5 criteria, for a person to be diagnosed with SLD they must have an IQ > 70 ([1], p. 69).

The Psychiatric Department of the “Papageorgiou” General Hospital adapted the WAIS-III in the Greek language, even though the scale has not been standardized in Greece. More specifically, the team of neuropsychologists translated the Verbal Scales of WAIS-III into the Greek language and used the Non-verbal Scales in their original version. The department used this adaptation for many different types of adults’ assessment in the hospital (e.g., neuropsychological, psychiatric, and learning assessment) [34,35].

### 2.2. Data Evaluation

The results of the overall evaluation were collected from “Papageorgiou” General Hospital’s SAP database system (Systems Applications and Products in Data Processing) [37] and the participants’ clinical and social histories. Data processing was performed using SPSS 25.

#### Statistical Methods

Statistical processing of the data collected for the present study was performed using the IBM SPSS Statistics software, Version 25.0 (Statistical Package for Social Science) [38]. The results of the research were presented using descriptive statistical techniques as well as statistical inference tools and were presented in the form of frequencies and relative frequencies. The Non-Parametric Pearson chi-square independence test was used to correlate variables identified by nominal variables. The Pearson chi-square test, also known as χ^2^-test, is a statistical test applied to determine if there is a significant difference between the expected frequencies and the observed frequencies in one or more variables. It was used to examine possible correlations between all the quality variables. The significant level for accepting or rejecting the respective case controls is in each case a = 0.05.

## 3. Results

### 3.1. Socio-Demographic Characteristics

The majority of the findings of the current study regarding the epidemiology, comorbidity, demographics and gender differences, predictors of education, and employment success of Greek adults with SLD are in agreement with other international studies. Figure 1 shows the epidemiological findings. More specifically, concerning the age of the participants, age was divided into four subcategories based on changes in a person’s life and the gradual transition from studies to work, to the possibility of family formation, until retirement.

### 3.2. Clinical Profile

#### 3.2.1. Mental Capacity

Mental capacity was measured with the WAIS-III/IV psychometric tool. The majority of the sample had average intelligence (68.5%), while 17.4% of the sample had low average, 5.8% high average, and 8.4% were diagnosed with borderline intelligence. Later analysis connected the mental capacity of the sample with their ability to live independently (Table A1). It was found that the percentage of people with normal or higher IQ scores that live alone was significantly higher than those with lower IQ (*p* < 0.001). People who did not live alone mentioned a history of previous learning difficulties during school age at a greater percentage than people who lived alone (*p* = 0.021) (Table A2). On the other hand, no statistical significance was found between the IQ levels and the rest of their demographic characteristics (Table A1).

#### 3.2.2. Diagnosis and Learning Profiles—SLD Types

Regarding diagnosis, four categories included all the different types of learning difficulties. In addition to SLD, the participants were diagnosed with Generalized Learning Disabilities (GLD), Language Difficulties, Mixed-type Learning Disorders, and Other types of comorbidity (Figure 2a). Specifically, adults with GLD have difficulties in all areas of learning, lower intellectual abilities, and significant impairment of social or adaptive functioning. The difference between GLD and SLD is that in the latter individuals have difficulties in one or more of the basic three areas of learning (reading, written language, and/or mathematics), whereas in GLD, they face difficulties in all areas of learning. Language Difficulties involve communication disorders in the processing and/or production of linguistic information, which can affect the overall learning process. As regards the mixed-type Learning Difficulties, the diagnosis is often unclear due to possible comorbidity with other neurodevelopmental disorders (e.g., ADHD, Language Difficulties, autism spectrum disorder, developmental coordination disorder, etc.) In the last category, the participants were diagnosed with comorbidity with a mental or psychiatric disorder (e.g., anxiety disorder, depressive disorder, etc.)

In addition, the current research studied the specific types of SLD, as formulated in DSM-5. The participants manifested learning difficulties in reading, written expression, or mathematics or two (or three) of those areas combined (“mixed”). It is important to emphasize that none of the adults presented learning difficulties only in the field of mathematics. Difficulties in this area arose only in conjunction with one of the other areas (“mixed”). Regarding the “other” type of SLD, this category included participants who have also been diagnosed with a comorbid disorder, such as ADHD, Language Difficulties or Specific Language Impairment, emotional/behavioral disorders, or anxiety disorders (Figure 2b). The statistically significant correlations are shown in Table A3 and Table A4.

#### 3.2.3. Possible Comorbidity and Previous Diagnosis

Several elements in terms of possible comorbidity and previous diagnosis were also evident (Table A5 and Table A6). A percentage of 90.2% stated that they faced difficulties in school, 43.6% had a previous diagnosis concerning learning difficulties in school, and 54.4% were diagnosed with possible comorbidity (Figure 3). In particular, people with reading difficulties showed higher levels of possible comorbidity (*p* < 0.001) (Table A3).

Additionally, the percentage of adults with other diagnoses, combined with GLD or Mixed type of diagnosis, was significantly higher than individuals who presented difficulties only in reading, writing, and/or mixed difficulties (*p* < 0.001) (Table A4). At the same time, participants with GLD and Mixed type were diagnosed with impaired developmental history (*p* < 0.001) and presented lower scores in intelligence scales than people with SLD (*p* < 0.001), as expected (Table A4). Furthermore, adults diagnosed with GLD or SLD showed a higher percentage of a previous diagnosis than those diagnosed with Mixed or other types of diagnosis (*p* < 0.001) (Table A4). However, individuals with SLD showed a lower percentage of comorbidity (*p* < 0.001) than those with GLD (Table A4).

Developmentally, participants with prior diagnosis, in their majority, were younger than those without a prior diagnosis (*p* < 0.001) (Table A6). In general, all kinds of diagnoses had a statistically significant correlation with the type of SLD detected (*p* < 0.001), and with the referral request reasons for assessment (*p* < 0.001) (Table A4). On the other hand, no significant association was found between diagnosis and demographics. The only case where the type of learning disorders and demographics were correlated was with regard to the participants’ occupation (Table A3).

#### 3.2.4. Impaired Educational History

Regarding the variable of educational history, the majority of the sample who mentioned impaired school history and had received a diagnosis earlier in life (*p* < 0.01) (Table A4), were of a younger (adult) age (*p* = 0.019) (Table A7). As expected, the percentage of the participants with an impaired developmental history, who studied in Secondary Education, was extremely high (*p* = 0.017), in contrast to those who managed to study in Postsecondary Education (*p* < 0.001) (Table A8 and Table A9).

Apart from the educational history of the participants, our interest concentrated on family and occupation status (Table A8). The majority of the adults with impaired school history were unmarried (*p* = 0.002) but did not live alone (*p* = 0.021) (Table A7). The percentage of participants who stated that they were single and had a prior diagnosis was significantly higher than those who were married, divorced, or those who lived with their partner (*p* = 0.022) (Table A6). Overall, impaired school history was found to have a statistically significant reliance on the participants’ professional/occupational profile (Table A7).

#### 3.2.5. Referral Request Reasons

The reason of request for assessment was related to age (*p* < 0.001), sex (*p* = 0.033), education (*p* < 0.001), and career (*p* = 0.038) of participants (Table A10). More precisely, 29.6% of the individuals requested an assessment for the renewal of their previous certificate of SLD and in order to receive a permanent exclusion from written exams. Other reasons included their participation in university entrance exams (Panhellenic Examination) (18.2%), placement exams or exams within the context of postsecondary education (17.6%), exams within the school context, technical college exams, or second-chance school exams (11.6%) and other personal reasons (23.0%).

Specifically, as mentioned above, the referral request reason for assessment was differentiated by the age of the participants (*p* < 0.001), since there was a lower percentage of younger adults that mentioned impaired educational history (Table A10). The request was further differentiated by sex, since males were referred for learning assessment at a higher percentage, for exams within the school context, technical colleges, second-chance schools, or renewal of their previous diagnosis (permanent exclusion from written exams) (*p* = 0.003) (Table A10 and Table A11). Interestingly, the percentage of participants over 31 years of age, who also had a previous diagnosis, was significantly lower than those up to 30 years old (*p* < 0.001) (Table A6). In addition, the findings show that IQ scores presented a significant correlation with the request (*p* < 0.001) (Table A1), whereas individuals with mixed type of diagnosis presented a higher percentage of possible comorbidity (*p* < 0.001), developmental history (*p* < 0.001), and the majority of them had received other types of diagnoses (*p* < 0.001) (Table A4).

#### 3.2.6. Age—Family Status—Occupation/Career

The participants with impaired educational history were mostly of a younger age, as opposed to those who did not face problems at school (*p* = 0.019) (Table A7 and Table A12). At the same time, the majority was married (*p* = 0.002) (Table A7). However, the percentage of unmarried adults with a prior diagnosis was significantly higher than those who were married and did not have a prior diagnosis (*p* = 0.022) (Table A13). Finally, participants with impaired educational history lived with other people at a higher percentage (*p* = 0.021) (Table A7).

## 4. Discussion

The overall findings of the study come to a total agreement with the basic diagnostic criteria for SLD internationally since the majority of the sample was found with normal intelligence levels along with persistent difficulties in one or more of the three basic academic domains (reading, writing, mathematics) [1,2,39]. Nevertheless, regarding the actual type of SLD, the fact that a very high percentage of the participants (42.5%) were found to be struggling with a “mixed type” of SLD, agrees with the new diagnostic classification of the DSM-5. This classification eliminates, to an extent, the previous categorization of SLD types and introduces the new broader term “Specific Learning Disorder”, which could include difficulties of various severity in the areas of reading, written expression, and mathematics, at the same time [1].

Regarding mathematics, it is necessary to consider the fact that in the present study none of the participants showed SLD exclusively in mathematics (also known as dyscalculia), but difficulties in this area arose only in conjunction with one of the other areas of SLD (reading and written expression). This result is impressive especially considering Koumoula’s study, in which the number of children in Greece who as students face difficulties in this area was 6.3% [40], and the fact that the prevalence rate worldwide for dyscalculia is around 5–10% [41].

The reason for this rather contradictory result—zero percent of SLD only in mathematics—is still not entirely clear, but it could be related to the fact that the specific research consisted of random individuals, namely, adults who voluntarily visited the OSDDL for evaluation, because of the difficulties they were facing in their daily life, their education and/or their work. This finding is, therefore, directly related to their referral request reasons. The manifestation of SLD, as presented in this study, was during the period in which the adults came in the OSDDL for evaluation. Hence, the results cannot be generalized to the whole population of Greece.

In addition, it is possible that some of the adults had been diagnosed with mathematical difficulties as children, and they had found alternative ways so that they could either learn to live with these difficulties or mask them. They most likely organized their lives and pursued occupations unrelated to mathematics. Moreover, the use of technology and mathematical applications in recent years certainly make it easier to learn how to deal, mask, or overcome some of these difficulties [42]. Although some of the participants in the study still had evidence of mathematical difficulties during their evaluation, they came in the OSDDL mainly because of difficulties they were still facing in the areas of reading and writing, comprehension difficulties or concentration problems, which inhibited their everyday lives, studies, or occupation. This can be interpreted by the fact that mathematics probably did not affect them (at least at present) in their everyday life, education, and/or work.

Moreover, the quite high level of possible comorbidity (54%) that was revealed through the analysis of our data, for adults with SLD—especially those experiencing reading difficulties—agrees with the recent research data of recent literature [30,43,44,45,46]. The current study’s findings confirmed that adults who experience a certain deficit in one particular learning domain frequently show deficits in other domains as well [4,9,16,17,21]. This finding provided further evidence about the complexity and the diffuse nature of the underlying cognitive mechanisms and the brain processing areas, which subserve reading, writing, and mathematical skills. The comorbidity “factor” interferes with the already existing issues regarding the establishment of a common clinical profile for adults with SLD. Therefore, all of these disorders/difficulties require a separate diagnostic procedure and the implementation of individualized intervention procedures and methods.

It should be mentioned that the overall picture of the results verified that SLD has a developmental nature and continues to influence several domains of a person’s life, even during adulthood. In particular, the analysis of the demographic characteristics of adults with SLD led to the following picture: The majority of the sample were male, single, young adults, up to 30 years of age, who had either graduated from Secondary Education or were still studying, most of them unemployed. Among the adults evaluated in the OSDDL Department, more males were diagnosed with SLD compared to females who had a lower rate of SLD. Males tend to exhibit more extrovert behavior, and as a result, they are more likely to refer for learning assessment at a certain point in their life. All the above findings indicate the ongoing struggle of this population to obtain academic qualifications in order to gain vocational rehabilitation, as well as their difficulty to create a family, possibly as a result of their lacking occupational status, their financial insecurity, and the emotional/self-esteem issues they usually encounter, due to their ongoing learning problems [4,14,31,47].

Additionally, there is a percentage of 56.4% of participants who had no previous diagnosis (Figure 3). Naturally, the previous diagnosis, where it existed, was not based on the DSM-5 criteria and especially, since the age of the participants increased. Therefore, this large percentage of undiagnosed SLD may also be due to the new diagnostic learning domains introduced with the DSM-5. Adults continuing in higher education are more likely to experience difficulties in comprehension and/or written expression. Within this 56.4%, there may be gifted adults who had managed to mask their deficiencies during their school years (giftedness might mask SLD) and those difficulties appeared more strongly with their entry into higher education [48]. This high level of undiagnosed SLDs can also be explained as a result of a lack of attention or non-recognition of SLD from the schools or from their parents, especially in cases of low socio-economic status.

Moreover, the fact that such a large number of people (56.4%) ask for a learning assessment as adults, for the first time in their lives, revealed the weaknesses that exist to this day, in relation to valid and early diagnosis of SLD. As a result, these individuals did not receive the appropriate intervention as children, which would have helped them as adults in their later life, outside school. Fortunately, in recent years, there seems to be a change of attitudes, since both parents and teachers more often turn to specialists when they suspect a difficulty in the child’s learning, in order to determine the appropriate intervention as earlier as possible.

The educational level of people with SLD had been the center of an ongoing debate among the scientific community. So far, many findings of previous studies have pointed out that students with SLD do not prefer higher education institutes, whereas, both international and Greek studies agreed that students with SLD were underrepresented in higher education institutions [7,31,49,50]. By contrast, a relatively high percentage of the present study’s SLD adult sample was enrolled or had completed higher education studies. The interpretation of this finding may be placed in the value given from the Greek society for a university degree as well as the fact that university education usually provides favorable prospects for employment and better income [13]. Moreover, the participants of the present study were adults seeking to achieve an academic target, referring themselves to learning assessment, mainly for educational purposes. Recent studies indicate an increased number of adults with SLD in postsecondary education [7,51,52,53,54]. As academic demands increase in young adulthood (university entrance exams, university term exams, or other types of exams), young adults with SLD seek a diagnosis for their learning deficits in order to receive accommodations. However, because of the difficulties they face, it is unknown how many of those who participate in university entrance exams (Panhellenic Exams) or those who are already enrolled at a university will be able to complete their studies on time or at all, which agrees with Vervelaki and Gritzepi’s survey [9].

In addition, the finding that the main reason for referral request was participation in several types of academic exams, once again validated the results of our previous study [31], which showed that adults in Greece seek learning assessment mainly due to socio-educational and socio-economic reasons. These findings may reflect the ten-year challenging period the Greek population experiences due to the economic crisis. These parameters also relate to later academic development, which usually leads to better vocational rehabilitation [4,9,18]. Overall, only 35.9% of the total sample of this study was employed (as private or civil servants, including those who studied and worked at the same time), and 64.1% were unemployed (including those who studied but did not work) (Figure 1). In the clinical interview, very few out of the 35.9% conceded that they had mentioned their learning difficulties to their employers, out of fear that this might affect their job performance. Most of them reported that they were ashamed to admit their difficulties to colleagues and their employer and that they were afraid of losing their job, which is in agreement with previous studies [2,12,14].

Moreover, as shown in Table A10, it was apparent that the referral request reason varied among different age groups of the participants. These findings verify that, even during adulthood, there are different age periods and in each one of them priorities differ (education, vocational rehabilitation, family, social life, etc.) [4]. More precisely, young adults, in particular, requested learning assessment, in order to continue their studies in secondary or higher education, with the aim of securing a better employment status in the future. Given the economic crisis in our country and the high levels of unemployment, it becomes obvious that this need is of great importance to young adults with SLD, especially in Greece.

With respect to the findings regarding the participants’ clinical profiles, it was found that from the 318 adults assessed, 235 (73.9%) were diagnosed with some type of SLD, while their main areas of learning difficulty were reading, written language, mathematics, or a combination of the above. Additionally, at an interpersonal level, different intelligence levels were detected (mainly within the normal IQ level). Finally, a variety of characteristics were recorded in all other life domains. Thus, all the findings concerning the clinical profile and the main type of diagnosis in the majority of participants, validate the enormous heterogeneity by which SLD can be manifested at an interpersonal level, as well as its developmental nature. This heterogeneity also involves different levels of intellectual capacity and different levels of adaptive behavior, both of which have a significant impact on the everyday overall functionality and social skills of the individual [4,16,20,23,24,55].

Regarding the social life of the participants, the data of the present research revealed that only 14.6% of those diagnosed with SLD were married, 3.8% were divorced and 81.6% had never been married. These figures showed that adults with SLD seem to encounter difficulties in finding long-term partners, as they lacked self-confidence due to avoidance of social interactions and situations where verbal communication was required for a reasonable period. They were also reluctant to disclose their difficulties and particularities to the person of interest, while, in several cases, their interpersonal relationships were either delayed, immature, or characterized by emotional instability. Previous studies have also shown that adults with SLD often do not fully understand the complex social relationships or the emotions that accompany them [10,18,19,23]. In other words, they appear insecure, without self-confidence, which makes it even more difficult to communicate with others.

An equally important aspect of this study was the fact that only 35.7% of those diagnosed with SLD lived independently, while 64.3% needed support from a family member in their daily life. These findings reinforced Hatton’s [21] and Bane’s et al. [22] claim that adults feel more secure in a familiar environment and find it difficult to abandon it. Moreover, in many cases, these people could not support themselves, either because they were not financially independent or because they required constant assistance in their daily lives [21,22].

Concluding, two positive and promising findings were the following: First, it seems that there is quite significant progress in the area of SLD diagnosis from the early years in Greece [7,9,31]. Second, as shown, people with “pure” SLD seem to be able to develop on both an academic as well as on a personal level, given that they are timely and accurately diagnosed [2,25,55]. Finally, the findings stress the need for timely career orientation [14].

## 5. Conclusions

Adults with SLD are a “sensitive” and unique population, who often refer to the diagnostic services requesting a learning assessment, either because they have nοt received an accurate and complete diagnosis of their learning difficulties earlier in life or because they had never been diagnosed before. As a result, they have never received the appropriate support and help they needed. The fact that the vast majority of the sample in this study was diagnosed with difficulties in at least two learning areas (reading and writing skills) verifies the complex nature of SLD and agrees with the DSM-5 diagnostic criteria [1].

Moreover, the various interpersonal characteristics, comorbidity issues, and different developmental backgrounds observed in the clinical, academic, personal, social, and occupational profiles of the participants highlight the enormous heterogeneity and the continuum that characterizes the SLD nature during adulthood.

Based on the results of the study and considering the limited research studies concerning adults with SLD (not only in Greece but internationally), it appears that there is an imperative need for further research and the construction of more sufficient tools for the assessment and diagnosis of SLD during adulthood. This will contribute not only to the improvement of the available diagnostic services but also to the improvement of the overall quality of life of this population.

Postsecondary education service systems should provide more appropriate study guidance to young adult students with SLD and should assist them to better cope with the challenge of academic tasks and exams, as well as with several life challenges. Besides, the need for timely information of parents and educators around the complex nature, diagnosis, and intervention policies of SLD seems is an important issue as well. More precisely, educators of all levels and parents should be provided with systematic information in order to raise their awareness for timely and proper diagnoses and provision of support for students with special educational needs and learning difficulties. It is commonly acknowledged that when detected at a young age, learning difficulties can be highly managed, which can lead to increased individual functionality and better quality of life. As the knowledge and awareness around this topic increases, it is believed that this will lead to a gradual decrease in the number of adult cases with undiagnosed SLD.

In conclusion, future research should be directed towards the development of a more comprehensive diagnostic system, with age-specific and more appropriate assessment tools that will take into account the developmental challenges and milestones in a series of life domains, in order to effectively assist this “vulnerable” population with their lifetime struggles.

## 6. Limitations of the Study and Future Research

Undoubtedly, the present study has some limitations. First, as the sample consisted of people who offered to participate, thus, many of the data collected through the clinical interviews and the self-reported questionnaires were based on the information shared by the participants themselves. It is very likely that these data contain subjectivity and inaccuracies and may reflect a need from the participant’s side to present a better life to the interviewers. Especially in the first years of the research, there were limited sources of information about these people. In many cases, the adults did not allow contact with a relative to ask relevant questions because they were ashamed of the difficulties they were facing. As a result, more objective sources of information were quite difficult to find.

Another limitation was the fact that participants were not assessed with psychological testing to further investigate more specific psychological and social aspects of their lives, in terms of the prevalence of additional issues, such as stress, aggression, delinquency, etc. Future research should also address these factors.

Furthermore, we believe that it would be extremely interesting to conduct a longitudinal study, to follow-up on the overall progress of these individuals and evaluate their current situation. Such a survey would allow further investigation regarding their progress over the years in several life-domains, i.e., whether they managed to complete their studies, find a job, or create a family.

In addition, a future research could be conducted to study the comorbidity issues with the SLD. According to DSM-5, there could be comorbidity between SLD and neurodevelopmental disorders, such as ADHD, Language Difficulties, autism spectrum disorder, developmental coordination disorder, etc., as well as between SLD and a mental or psychiatric disorder (e.g., anxiety disorder, depressive disorder, etc.)

Further analysis and utilization of the study findings can lead to significant benefits:(a)Regarding the detailed profiling of the clinical characteristics of adults with SLD in several life areas.(b)In terms of the construction of more appropriate, age-specific tools for a complete and multi-faceted assessment.

Finally, it could offer essential recommendations, both in terms of prevention, as well as at an intervention/assistance level, in order to effectively assist adults with SLD with the difficulties they face in several domains of their daily lives.

## Figures and Tables

**Figure 1 brainsci-11-00602-f001:**
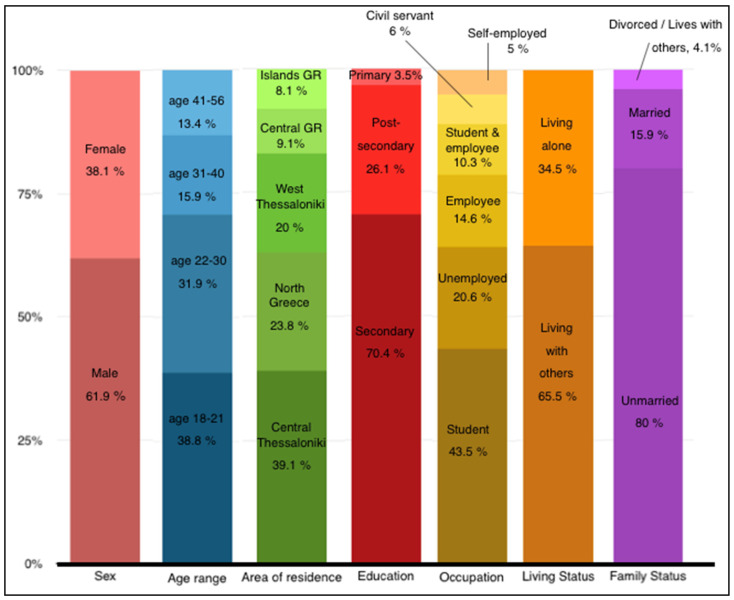
Socio-demographic characteristics of the participants that were assessed for SLD at the OSDDL of the 1st Psychiatric Clinic, “Papageorgiou” General Hospital of Thessaloniki, Greece.

**Figure 2 brainsci-11-00602-f002:**
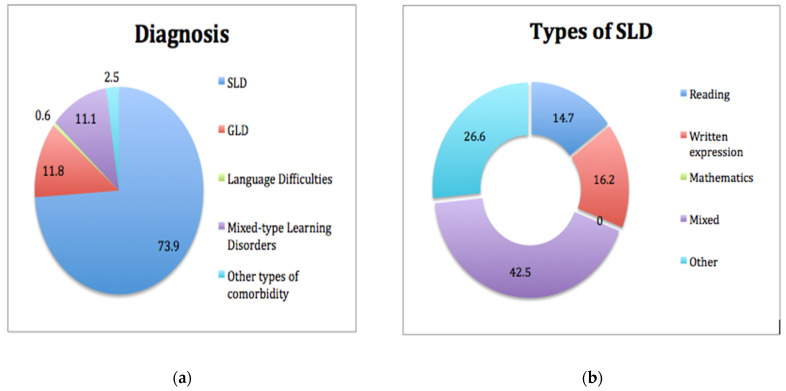
(**a**) Diagnosis of the assessed adults at the OSDDL. SLD = Specific Learning Disorder, GLD = Generalized Learning Disabilities, (**b**) Learning profiles of the participants that were diagnosed exclusively with SLD.

**Figure 3 brainsci-11-00602-f003:**
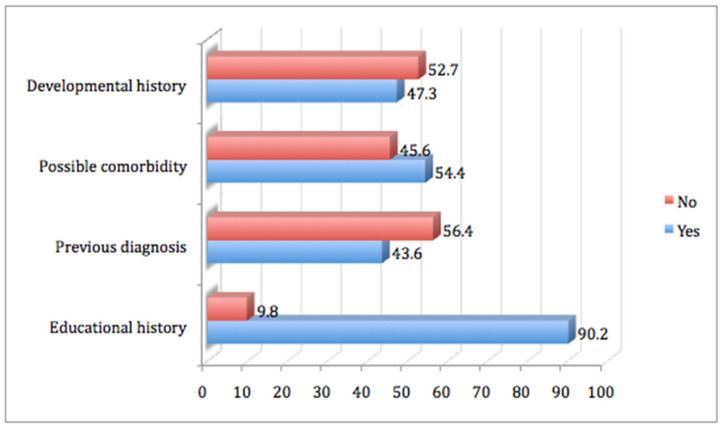
Clinical findings per diagnostic categories of the participants that were assessed at the OSDDL.

## Data Availability

The data presented in this study are available on request from the corresponding author due to privacy issues. The data are not publicly available due to privacy.

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
