# Peer review of "Clinical Profiles and Socio-Demographic Characteristics of Adults with Specific Learning Disorder in Northern Greece"

_brainsci, 2021, doi:10.3390/brainsci11050602_

Round 1
Reviewer 1 Report
It was a pleasure reading the manuscript focusing on the profiles and socio-demographic characteristics of adults with SLD and on the associations between the various included variables in this study in relation to everyday functioning life-skills of the adults with SLD (indications). Notwithstanding the added value of this perspective, especially addressing the adults group with SLD, the manuscript, as it stands now, needs some points of clarification, elaboration and justification. I will address the issues to care of in order to improve the manuscript.
- I really congratulate the authors using the DSM-V and elaborating on the broader definition of SLD within the DSM-V compared to the earlier version. However, the variable ‘previous diagnoses’ will most certainly not derive from the DSM-V (and the probability rises with increasing age of the participants). Hence, undiagnosed SLD’s (on page 5 of the manuscript being depicted as large numbers of students with undiagnosed SLD, however, without any reference) might be due to the new diagnostic domains of learning, i.e. reading comprehension and some new aspects of written language skills. Students embarking in higher education are more prone to be confronted with reading comprehension problems and written language skills, increasing the number of undiagnosed persons with SLD (DSM-V). It is recommended to elaborate on the issue of undiagnosed SLD’s as articulated in the introduction: the amount of participants in this sample being diagnosed earlier with (S)LD is more than 43 percent (with comorbidity is even higher), some of the LD’s might have been masked due to high abilities (for masking effects and a review on gifted students with SLD: see Beckmann & Minnaert, 2018, Frontiers in Psychology: Noncognitive characteristics of gifted students with learning disabilities: An in-depth systematic review). This masking effect also contributes why operational definition of SLD on adults is far more complicated compared to primary education school children, although masking (giftedness might mask SLD, or giftedness and SLD mask each other) might start already on a young age. Besides, schools and parents (especially of lower socio-economic status) might have not recognized these (S)LD’s either. Please problematize the number of undiagnosed SLD’s more in depth.
- Given the number of school children in Greece with mathematical skills (see e.g. Koumoula, et al., 2004, Journal of Learning Disabilities: An Epidemiological Study of Number Processing and Mental Calculation in Greek Schoolchildren), it seems not probable that none of the participants might have SLD for mathematics alone. Worldwide the prevalence of dyscalculia is 3-6%, so why is this completely absent in this study? The absence is even not reflected upon and in the text (in multiple sentences) the unique difficulties in mathematics are mentioned while the prevalence in the sample is zero percent. Please justify this statement and elaborate at least in the discussion why this is zero. As the publication in 2004 mentions that a large group of children has persistent difficulties in number processing abilities, how could the percentage here be zero. This issue questions the representativeness of your sample.
- With the use of chi-square tests, we look at the relationship between discrete variables. Correlations are preserved for looking at the relationship between (assumed) continuous variables. Change the wording ‘correlations’ in the heading of the Tables into ‘relationship’.
- Not all relationships are explicitly retrievable in the Tables. The relationship between Mental capacity and Educational history is not to be found in Table 1. It might be non-significant, as briefly mentioned in section 3.2.1, but for replication and/or meta-analytical purposes mentioning these statistics is of utmost importance. In section 3.2.4., Impaired developmental history is the heading, but most of the information is related to impaired school history. Besides, the nonsignificant relationships of impaired developmental history with Family status and with Occupation is not mentioned at all (Table A7). This is inconsistent with the non-retrievable relationship mentioned at the start of point 4.
- Underscore the comorbidity issues with SLD as implication for further research.
- Typographical errors: Histor in section 3.2.4 (should be history); Dysxlexia in reference 19 (should be Dyslexia).
I strongly hope all the comments given are helpful to upgrade the manuscript accordingly.
Author Response
"Please see the attachment."

Reviewer 2 Report
Dear Authors,
I appreciated your research article.
The Introduction is well structured.
The methods section is clear and the results are presented in a good manner. However, I suggest you to modify the lyout of the Figure 3 (more professional).
Discussion and conclusions are exhaustive.
Author Response
"Please see the attachment."
